# Patient-Derived Tumor Xenograft Models: Toward the Establishment of Precision Cancer Medicine

**DOI:** 10.3390/jpm10030064

**Published:** 2020-07-18

**Authors:** Taichiro Goto

**Affiliations:** Lung Cancer and Respiratory Disease Center, Yamanashi Central Hospital, Kofu, Yamanashi 4008506, Japan; taichiro@1997.jukuin.keio.ac.jp; Tel.: +81-55-253-7111

**Keywords:** patient-derived tumor xenograft (PDX), anti-cancer drug development, immunodeficient mice, precision medicine

## Abstract

Patient-derived xenografts (PDXs) describe models involving the implantation of patient-derived tumor tissue into immunodeficient mice. Compared with conventional preclinical models involving the implantation of cancer cell lines into mice, PDXs can be characterized by the preservation of tumor heterogeneity, and the tumor microenvironment (including stroma/vasculature) more closely resembles that in patients. Consequently, the use of PDX models has improved the predictability of clinical therapeutic responses to 80% or greater, compared with approximately 5% for existing models. In the future, molecular biological analyses, omics analyses, and other experiments will be conducted using recently prepared PDX models under the strong expectation that the analysis of cancer pathophysiology, stem cells, and novel treatment targets and biomarkers will be improved, thereby promoting drug development. This review outlines the methods for preparing PDX models, advances in cancer research using PDX mice, and perspectives for the establishment of precision cancer medicine within the framework of personalized cancer medicine.

## 1. Introduction

Tumor cells have high proliferative activity, and they actively undergo DNA synthesis. DNA damage is more likely to occur in tumor cells than in normal cells, and this damage can lead to the loss of cell viability [1]. By closely examining these features of tumor cells, compounds with the following mechanisms of action have been developed as anti-cancer drugs: (1) direct damage of DNA through adduct formation; (2) suppression of DNA synthesis through the inhibition of nucleic acid metabolism or DNA–protein complex formation; (3) suppression of cell division through the inhibition of the function of proteins involved in cell division (e.g., tubulin). Numerous standard methods of tumor treatment, each of which is designed to suppress tumor cell proliferation and induce the loss of tumor cell viability, have been established by combining two or more of these compounds [2,3,4,5]. These treatment methods are based on the concept “one size fits all,” which aims to use the same treatment method for many patients [6]. Although these methods have displayed some level of therapeutic efficacy, their effects are sometimes insufficient because of adverse reactions or drug resistance. For this reason, treatment methods matching individual patients, i.e., “personalized treatment”, have been explored [1,6].

Following advances in methods for studying molecular biology, thorough gene analysis of tumor cells has been conducted in recent years, leading to the detection of driver mutations involved in the acquisition of traits promoting tumor cell survival (e.g., enhanced cell proliferation, resistance to apoptosis) [7,8,9]. Furthermore, molecular targeted drugs, which aim to specifically eradicate tumor cells, have been newly developed by targeting these gene mutations [8,9,10,11]. These drugs are transforming cancer therapy, as observed for the HER2 inhibitor trastuzumab in breast cancer, c-Kit inhibitor imatinib in chronic myelogenous leukemia and gastrointestinal stromal tumor, and epidermal growth factor receptor inhibitor cetuximab in colorectal cancer [8,12]. Furthermore, “immune checkpoint blockade” therapy designed to eradicate tumor cells through the reactivation of tumor-suppressed immune function has proven effective in select patients. Drugs in this class, such as ipilimumab (anti-CTLA-4 antibody) and pembrolizumab (anti-PD-1 antibody), have already been introduced clinically [12,13,14,15,16].

Following such remarkable advances in the field of tumor treatment, active research has been conducted at a global scale to facilitate the development of additional molecular targeted drugs. However, in the research and development of molecular targeted drugs, the results of non-clinical studies rarely predict clinical efficacy. This paradox is largely attributable to the lack of appropriate non-clinical models reflecting the diversity and complexity of tumors [17,18]. In other words, the existing DNA-damaging anti-cancer drugs are based on a relatively common and simple mechanism, namely the high proliferative activity of tumor cells. These drugs have been analyzed using models involving the constant proliferation of tumor cells [14]. However, the development of molecular targeted drugs, which target specific molecules in diverse and complex tumors, requires a model that permits the appropriate expression and function of the target molecule in tumor cells [19,20]. In this context, the xenograft model, which involves the implantation of cultured tumor cell lines established from tumor tissue into immunodeficient mice, has been often used as an in vivo model for cancer research [19,20]. As is the case for in vitro models, constant tumor cell proliferation is maintained in xenograft models, and the validity of anti-cancer drugs based on the results of non-clinical studies has been assured to some extent [21]. Furthermore, the correlation of the outcomes of preclinical studies using xenograft models of cell lines possessing certain driver mutations with clinical efficacy is known [19,22]. However, because cultured cell lines consist only of specific tumor cells adapted to culture conditions that differ markedly from the in vivo environment, xenograft models of cultured cells are not considered, at present, to reflect the diversity and complexity of tumors [19,22]. One model type expected to resolve this open issue is the patient-derived xenograft (PDX) model, which involves the direct implantation of tumor tissue into immunodeficient mice without in vitro incubation (Figure 1) [18]. In the PDX model, the molecular, genetic, and histological characteristics of tumors are preserved, and it is also possible to compare PDX models among multiple cases [23,24,25,26,27]. Thus, this model is expected to be immediately applicable to research because it reflects the diversity and complexity of tumors [23,25,28]. Furthermore, it is possible to confirm the findings of PDX models using the patient tumor tissue from which the PDX model originated. This model type is thus expected to represent a tool used for translational research, serving as a bridge between non-clinical and clinical studies [18,28,29].

The development of novel anti-cancer drugs is an important task for prolonging the survival of patients with cancer and curing cancer itself. In recent years, active efforts have been made toward the development of new types of anti-cancer drugs (e.g., molecular targeted drugs, immune checkpoint inhibitors) in addition to the existing cytotoxic anti-cancer drugs. However, the probability that a novel anti-cancer drug will reach clinical use after the successful completion of clinical trials (Phases I through III) is as low as 5%. It is particularly common that the investigational new drugs cannot enter Phase III development after Phase II trials. This issue slows the pace of new drug development, thereby markedly increasing the expense of drug development [30]. In other words, many drugs that exhibit efficacy in preclinical studies fail to demonstrate sufficient efficacy in patients.

Compared with cell line-derived xenograft models, PDX models are expected to feature high predictability of therapeutic efficacy while preserving the inhomogeneity of the patient’s tumor [31]. Large-scale PDX libraries are currently being established in Europe and the USA. In 2016, the US National Cancer Institute announced plans to switch its anti-cancer drug screening system from the “NCI-60 Human Tumor Cell Lines Screen” to PDX-based models [32]. The precise prediction of responses to treatment cannot be achieved by simply measuring the expression of proteins or mutation of genes. Furthermore, when treatment in individual patients is decided on the basis of genomic information, beneficial effects are achieved in only a limited number of patients. In addition, the identification of biomarkers based on information about the molecular background is time-consuming and insufficient for the speed of anti-cancer drug development. Under such circumstances, the PDX model is anticipated as a means for supplementing or replacing molecular biomarkers.

## 2. Development and Modification of Immunodeficient Animal Models

To create a PDX model, immunodeficient mice are indispensable. Human tumors will initiate a graft versus host reaction, leading to rejection in immune-competent mice. In practice, the idea of the PDX model was triggered by the discovery of immunodeficient mice. The first generation of immunodeficient mice consisted of nude mice lacking T cells [33]. These mice were discovered early in the 1960s. Dr. Rygaard (Denmark) reported a nude mouse lacking the thymus and T lymphocytes, and it displayed a defect in T-cell-mediated immune responses and antibody formation that requires helper T cells [34]. The use of this mouse stimulated remarkable progress in basic research on immunity and cancer [35]. In addition, they remain an important resource for PDX establishment because nude mice have benefits including a relatively high engraftment ratios of gastrointestinal tumors, easy observation of subcutaneous tumors due to lack of hair, and relatively low price [31,36,37]. Later, a wide variety of immunodeficient mice were developed and modified [38]. Table 1 summarizes the lineage and characteristics of immunodeficient mice used specifically for PDX models.

Via repeated efforts to improve the efficacy of grafting, severe combined immune deficiency (SCID) mice lacking T and B cells were developed, making it possible for the first time to successfully graft human blood cells [39,40,41]. This was initially termed the “SCID-hu system”. However, sufficient grafting required fetal tissue, and the manifestation of acquired immunity (leakiness) occurred over time in theoretically immunodeficient mice [42]. Thus, the graft survival rate was not sufficient in this model. In the 1990s, SCID mice were crossbred with non-obese diabetic (NOD) mice, yielding NOD/SCID mice with composite immunodeficiency (e.g., T/B cell defect + NK cell malfunction) [43,44,45,46]. NOD/SCID mice have a markedly improved human hematopoietic cell graft survival rate, and they remain in extensive use. However, NOD/SCID mice have several shortcomings, including the lack of T cell graft survival, the absence of long-term observation (because of the high incidence of thymoma and a short lifespan), and the lack of graft survival for stem cell systems other than the hematopoietic system.

To further improve the graft survival rate, in the 2000s, a new trait (common gamma chain knockout) was introduced into NOD/SCID mice, yielding NOG mice [47,48]. NOG mice were created by crossbreeding NOD-*scid* (NOD/Shi-*scid*) mice with IL-2Rγc^null^ mice, and they exhibit composite immunodeficiency (T, B, and NK cell defects and dendrocyte/macrophage malfunction; Table 1) [38,49,50]. Because of these characteristics, NOG mice are considered the best immunodeficient animals for human tissue graft transplantation [18,50]. In essence, engraftment ratios are higher in more immunocompromised mice (Nude < SCID < NOD/ SCID < NOG) (Table 1) [36].

## 3. Creation of PDX Models

A PDX model is created by grafting a patient-derived tumor sample into immunodeficient mice (Figure 1) [37,51]. Usually, tumor growth begins within several months after grafting (F0). At that time, part of the graft is used for genetic analysis, such as whole-exome sequencing, RNA sequencing, and copy number alteration, to analyze the genetic characteristics of the tumor, and another part of the graft is stored in the PDX tissue bank. Part of the tissue is grafted into immunodeficient mice (F1), and the tumor after proliferation is frozen in a large quantity. If these tumor samples are grafted simultaneously into numerous mice and candidate drugs are administered to the mice, it is possible to screen and identify drugs that are effective against the patient’s tumor. If the results of gene analysis are compared with the clinical data, it is possible to cultivate the path for personalized treatment. With the PDX model, tumor growth in vivo in mice accelerates as passaging is repeated. In the F3 generation, the patient-derived sample is approximately identical to the genetic expression profile. Thereafter, the genetic profile changes with further passaging [18]. For this reason, it has been recommended to use tumor of F3 or earlier generations in PDX model-based evaluation.

PDX tumors possess the genetic characteristics and tumor heterogeneity that are more similar to the patient’s derived tumor than to the tumor cell line. These tumors contain patient-derived cells such as stromal cells/cancer-associated fibroblast and tumor-associated macrophages before they are gradually replaced by mouse cells as the passage increases. PDX models, especially in the early passages, are therefore expected to be applicable to clinical and pathophysiological analyses of tumor and anti-tumor drug development because the models most closely resemble the clinical environment. PDX models have already been created for many cancer types. Reports are available concerning their creation for solid cancers (e.g., breast cancer, lung cancer, pancreatic cancer, colorectal cancer, melanoma, head and neck cancer, prostate cancer, renal cell carcinoma, glioblastoma, ovarian cancer) and blood tumors (e.g., leukemia, lymphoma) [18].

Following the recent development of intensely immunodeficient mice presenting with various types of immunodeficiency, the grafting efficiency of PDX models has been remarkably improved (Table 1). However, the long-term rearing of intensely immunodeficient mice requires an SPF (specific pathogen free) setting, and the breeding of such mice is difficult. It is therefore desirable to use mouse strains that are suitable for the target type of tumor by precisely assessing the advantages and shortcomings of individual mice. It must be considered that 3–6 months are usually required to establish a PDX model.

The graft survival rate of PDX models varies depending on the type of tumor involved (Table 2). Colorectal and pancreatic cancers have high graft survival rates, and a favorable outcome to some extent may be expected even when nude mice are used. Conversely, ultra-intensely immunodeficient mice such as NOG, NSG (NOD/Scid/IL2Rγ-null), and NOJ (NOD/Scid/Jak3-null) mice are required to establish PDX models of hematopoietic tumors [52]. The grafting efficiency usually tends to be higher in mice with more intense immunodeficiency, but mice with intense immunodeficiency are more difficult to rear. Furthermore, the probability of the successful establishment of PDX lesions is higher for metastatic foci than for the primary lesion, and this tendency is more marked for tumors with greater malignant potential [53]. Regarding the shape of the tumor used for this model, a square tissue section of several millimeters in size, excised from the patient’s tumor via biopsy or surgery, is usually used. In recent years, reports have described the creation of PDXs using circulating tumor cells or bodily fluids (e.g., cancerous hydrothorax, cancerous ascites) [54]. Regarding the recipient site, subcutaneous tissue is generally used because subcutaneous grafting is simpler and it permits the easy evaluation of tumor growth after grafting. However, the efficiency of subcutaneous grafting is low for breast and prostate cancers. For this reason, breast cancer is grafted into the mammary glands of female mice to enable the efficient creation of PDXs, and prostate cancer is grafted into the prostate glands of male mice (orthotopic implantation) [55]. It is also known that the grafting efficiency of hormone-dependent tumors (e.g., breast cancer, prostate cancer) increases if human hormones are replenished [18,53]. Thus, four essential elements for the establishment of PDX models are as follows: (1) properties of the tumor (primary lesion/metastatic foci or surgical specimen/biopsy specimen/humoral cells); (2) selection of recipient mice; (3) recipient site; (4) replenishment based on tumor characteristics (hormone treatment).

In practice, the graft survival rate varies greatly depending on the tumor type. High graft survival rates (80% or higher) have been reported for malignant melanoma and colorectal cancer, whereas the rate is as low as approximately 30% for breast cancer (the mean graft survival rate for 18 tumor types was approximately 50%; Table 2) [56,57]. Furthermore, in the case of triple-negative breast cancer, the graft survival rate following orthotopic implantation is 60–86% (more than twice the rate after subcutaneous implantation). This also suggests the importance of recipient site selection.

In recent years, active efforts have begun to be made toward the development of new methods of cancer treatment focusing on human immunity (e.g., immune checkpoint inhibitors). The PDX model uses immunodeficient mice, i.e., mice with markedly compromised immune function. Therefore, the balance of hematopoietic and immune cells remains different from that in humans and theoretically, the immune cell–tumor cell interaction may be totally missing in this model. If these drugs are administered to immunodeficient mice, then reactions of immunocompetent cells differing from those observed in humans are anticipated. These issues require attention for the development of a patient-similar immune response PDX model. For this reason, humanized mice (intensely immunodeficient mice implanted with human immunocompetent cells or umbilical cord-derived hematopoietic stem cells) are sometimes used to create PDX models. In some studies, humanized mice were created using immunocompetent cells derived from patients with cancer, and these mice were implanted with the patient’s tumor to create PDXs and evaluate drug efficacy [58,59]. However, this approach is not yet extensively applicable because of problems as follows: (1) there is no model completely reflecting the human immune system; (2) intense host rejection occurs frequently, making long-term evaluation difficult; and (3) the costs are high [38].

## 4. Prediction of Cancer Response to Treatment Using the PDX Model

Large-scale PDX libraries are being established in Europe and the USA and utilized for drug development and biomarker screening. In 2013, “EurOPDX” was organized by 16 universities and public institutions in Europe. To date, more than 1500 PDX types, including rare cancers, have been established [56]. Furthermore, the Jackson Institute (USA) has begun to publicize the genetic and histopathological information of more than 450 types of PDXs established on its website. At the same time, the institute has initiated the industrial utilization of PDXs by commercializing various PDX mice.

The greatest advantage of PDX use is the high predictability of treatment response. When the efficacy of 129 drugs was evaluated using 92 PDX types, the response rate evaluated using PDX models had 87% agreement with the clinical response rate (112/129), and the variance depending on the site of the primary lesion was also small (81–100%) [57]. In another study designed to compare the efficacy of treatment using approximately 1000 types of PDXs with the clinical efficacy, the rate of agreement was as high as 66% [60]. This agreement rate was excellent compared with that of conventional tumor cell grafting models (approximately 5%). If drugs targeted for clinical development and the subjects anticipated to respond to them can be selected efficiently using PDX models in preclinical studies, the mental/physical stress on clinical trial participants and the cost of clinical development will be reduced. Furthermore, because PDX models enable tumor collection over time (e.g., before and after drug efficacy evaluation), PDXs may contribute to clarifying the pharmacokinetics, response, and resistance development mechanisms of tumors.

PDX models are also expected to play important roles in the development of treatments for rare cancers. As the definition suggests, the number of patients with rare cancers is extremely small. This feature is usually considered disadvantageous from the viewpoints of diagnosis and treatment. Furthermore, because the collection of clinical specimens and implementation of large-scale clinical trials are difficult for rare cancers, the development of treatments generally does not proceed smoothly. Therefore, the creation of PDX models using specimens from patients with rare cancers may permit the collection of samples needed for drug efficacy evaluation and pathophysiological analysis. In addition, this strategy will enable the confirmation of proof of concept during preclinical or early clinical studies, thus promoting remarkable advances in the development of treatments and clarification of the mechanisms of carcinogenesis (e.g., genetic factors).

## 5. Cancer Stem Cells and PDXs

Previously, cancer was considered to be a group of single cells that deviated from the normal mechanism for preserving the living body, resulting in permanent proliferation. However, it has been increasingly recognized that a group containing a small number of cells called “cancer stem cells” is present in both blood tumors and solid cancers [61]. According to this cancer stem cell hypothesis, stem cells similar to normal tissue are also present in tumor tissue, and they have the potential for self-renewal, accompanied by the ability to form a tumor similar to the original tumor tissue despite being small in number [62]. Cancer stem cells are considered to maintain stemness for a long period, similarly to tissue stem cells. They divide more slowly than daughter cells, and thus, they are resistant to conventional anti-cancer drugs or radiotherapy (Figure 2) [63]. Furthermore, it has been suggested that these cells exhibit relatively strong resistance to molecular targeted drugs [64]. Therefore, the risk for recurrence or metastasis is considered high if cancer stem cells are not eradicated, even in cases in which the tumor is not clinically detectable after treatment. Furthermore, recurrent or metastatic cancer usually has high resistance to drugs, and therefore, it has been highlighted as a major factor for poor prognosis. Under such circumstances, treatment methods targeting cancer stem cells are anticipated as a promising therapeutic strategy.

It has been reported that a specific subpopulation within a tumor is more likely to form a tumor than the other subpopulations when grafted into mice [61]. Following this finding, the concept of cancer stem cells became widely accepted. To reveal that specific cells have the nature of stem cells in a study of cancer stem cells, it is necessary to demonstrate in vivo that a tumor composed of heterogeneous cell groups can be formed. When such an analysis is conducted, PDX are more informative than cell lines, which have been maintained for a long periods in two dimensions, on a non-natural surface, which may have led to genetic drift and selection [31]. Because cancer stem cells are present in tumors in extremely small quantities, there may be cases in which the tumor is too small to secure the quantity of stem cells needed for analysis via flow cytometric sorting. In such cases, new PDXs having undergone a few rounds of passaging are sometimes used [65]. If the nature of cancer stem cells is further clarified using PDX models, then the development of effective treatments, either selectively targeting cancer stem cells or simultaneously targeting both cancer stem cells and other cancer cells, will be facilitated.

## 6. Open Issues Related to PDX Models

PDX models are considered promising tools for tumor research, but several open issues have been identified. PDX models cannot be established from all patient-derived tumors. It has been stated that the models are difficult to establish from some tumor tissues [18,20]. Furthermore, substantial resources (e.g., money, time, labor) are needed to create PDX models [18,20]. To extensively use PDX models in tumor research, it is indispensable to improve the efficiency of model creation by using greater amounts of tumor tissue [18,19,20].

If genetic analysis and drug sensitivity testing are conducted and the tumor characteristics and clinical information are analyzed simultaneously, established PDX models, as well as the databases constructed from the collected data and information, are expected to be useful in the personalization of treatment corresponding to the characteristics of tumors in individual patients. The EurOPDX and Jackson Institute databases contain large numbers of samples, and they have started to make samples publicly available [52,56]. Some pharmaceutical companies are also creating their own PDX libraries. Novartis reported the results of drug screening using more than 1000 PDX types [60]. The tumor characteristics differ by ethnicity and region. For this reason, it is desirable that such large-scale libraries are established globally.

As mentioned previously, a considerably large number of PDXs have already been established and utilized in various studies (e.g., drug efficacy evaluation) and the promotion of personalized treatment. However, the methods and evaluation of PDX creation and studies using PDX have not yet been sufficiently standardized. For example, the appropriate frequency of PDX passaging differs among investigators. “EurOPDX” recommends the use of PDXs after five or fewer passages for studies such as drug efficacy evaluation because the tumor stroma derived from the patient is increasingly replaced with mouse stroma with increasing passaging, possibly causing discrepancy of the clinical interactions between tumor cells and the stroma [56]. A study by the Broad Institute demonstrated that genomic mutations present in patients’ tumors are lost with increasing passaging, resulting in the loss of the genetic characteristics of the tumor [66]. Furthermore, it is plausible that dynamic changes of genes occur during the course of growth and infiltration of patients’ tumors, occasionally leading to the appearance of mutations not present in PDXs. For this reason, it is difficult to set certain conditions for each PDX unilaterally. It appears essential to store patients’ specimens and the tissue/specimens obtained during passaging in close linkage with data including treatment history, genomic information, and other data collected from various studies (e.g., omics analysis). In addition, it is critical to use the PDX for each study after sufficient assessment concerning whether the model satisfies the requirements for a given setting of use.

The maintenance of PDX libraries results in the accumulation of large amounts of data from clinical practice and studies, and these data should be maintained with high levels of privacy. According to the current main practice in Europe and the USA, the collected data are made public on websites and other media, after anonymization, to permit the free use by researchers and clinicians. However, the control and disposal of such data requires significant labor and financial expenditure. It is therefore difficult for an individual or single institution to establish and maintain a PDX library. It is desirable to form a consortium of multiple institutions, clearly defining to whom the PDX, as well as the patent and financial rights related to its research, should belong at the stage of planning.

Furthermore, crossreactivity between human and mouse proteins (e.g., humoral factors) can occur in PDX models. Regarding the genetic homology between frequently used experimental animals and humans, the homology between mice and humans has been reported at approximately 92% (versus 99.9% between humans). Thus, even proteins with the same function can differ slightly in terms of amino acid structure between humans and mice, leading to the lack of binding between a ligand and its receptor and the absence of signal transduction [67]. To resolve this problem, attempts have been made to achieve a more faithful reproduction of diseases by introducing human cytokine and HLA genes into various immunodeficient mice and creating next-generation immunodeficient mice capable of producing human proteins (e.g., cytokines) [68,69,70].

## 7. Application to Personalized Treatment

In the case of cancer, heterogeneous cell groups are present within a single tumor, and furthermore, tumors located at the same site exhibit many differences between patients [71,72]. For this reason, the need for personalized treatment is being emphasized. Identification of the gene mutation serving as a driver via target or exome sequencing provides a useful tool for predicting an effective treatment method in silico, and this strategy is expected to be adopted actively in the age of personalized treatment [73,74,75]. If the prediction of treatment methods using bioinformatics is combined with the evaluation of responses to drugs in vivo using PDX models, integrated personalized treatment will be enabled. The PDX model created using tumor tissue derived from individual patients can supply information regarding the response of tumors to various drugs, thus enabling evaluation of the efficacy of treatments prior to clinical use (Figure 3). Thus, PDXs are expected to contribute to advancing the personalization of treatment.

Meanwhile, PDX models can also take significant time to create, which may pose a challenge to patients with advanced stages of cancer. For example, growth rates of prostate cancer PDX are slow, needing many months to generate models [76]. The time to initial growth is reported to be from four up to over 12 months, and time from implantation to initial growth of secondary passage ranges from 6 to 36 weeks, partly due to the differences in androgen levels between human and mouse [26,77]. This time-consuming process is clearly beyond the period that would be useful to define the best treatment modality. If problems such as the low graft survival rate and the time required for tumor formation can be resolved, PDX models will be extremely helpful in tailoring treatment to individual patients.

## 8. Conclusions

Following the recent development of intensely immunodeficient mice that permit the grafting, proliferation, and differentiation of human cells, it is currently possible to create PDXs of various tumors. PDX models faithfully reproduce human tumors. If drug screening is extensively conducted, then the number of new drug candidates that fail to proceed beyond phase II development may be reduced, thereby improving the efficiency of anti-cancer drug development. This will undoubtedly serve as an extremely valuable tool for translational research. In the near future, further modification of intensely immunodeficient mice and the enrichment of PDX libraries are expected to stimulate further advances in the application of PDX models to precision cancer medicine.

## Figures and Tables

**Figure 1 jpm-10-00064-f001:**
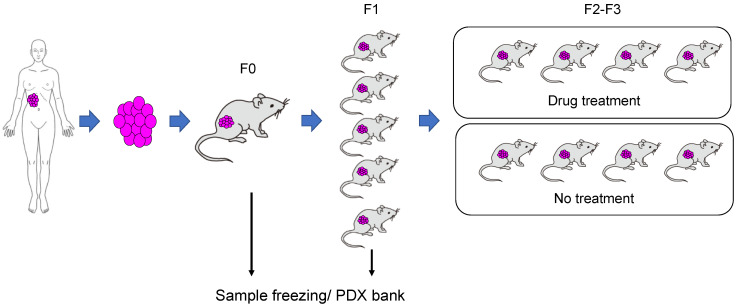
The grafting of patient-derived tumor cells into intensely immunodeficient mice leads to tumor growth in the mice within several months. The expanded tumor is excised. Part of the tumor is frozen, and the other part is grafted into intensely immunodeficient mice. These tumors are grafted again into intensely immunodeficient mice to conduct pathophysiological studies and evaluate drug efficacy. If the frozen patient-derived xenograft (PDX) tumor is registered with the PDX bank and a database of genetic analysis data and drug sensitivity data is created, it is expected to facilitate precision cancer medicine corresponding to the characteristics of the tumor in a given case.

**Figure 2 jpm-10-00064-f002:**
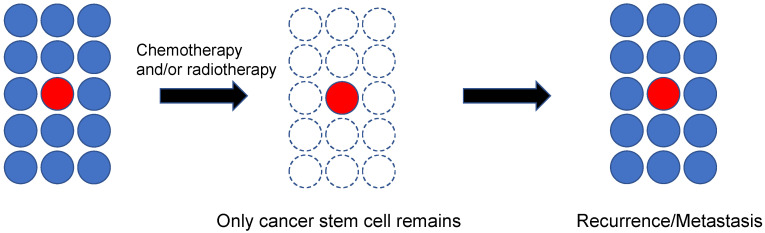
Schematic diagram of cancer stem cells responsible for recurrence/metastasis. Cancer stem cells are likely to remain after treatment because they are resistant to anti-cancer drugs or radiotherapy. They are considered to serve as a cause for recurrence/metastasis.

**Figure 3 jpm-10-00064-f003:**
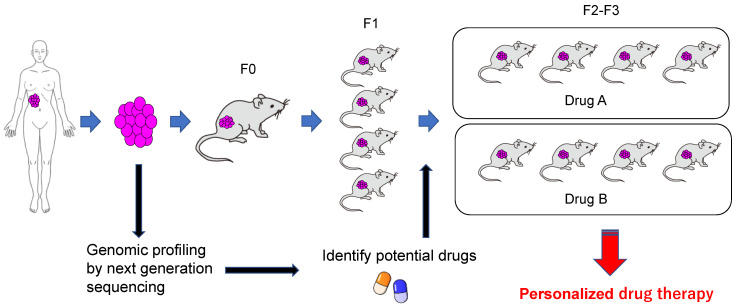
Grafting of patient-derived tumor cells into intensely immunodeficient mice leads to tumor growth within several months. The expanded tumor is excised. Part of the tumor is frozen, and the other part is again grafted into intensely immunodeficient mice. Multiple candidate drugs are selected corresponding to the genetic profile of a given tumor, and each drug is administered to the PDX mice to evaluate its efficacy. If drug sensitivity data collected using these are utilized for patients, then precision cancer medicine may become possible.

**Table 1 jpm-10-00064-t001:** Development and characteristics of immunodeficient mice used in animal models.

Characteristic	Nude	SCID	NOD/SCID	NOG
Reporting year	1966	1983	1995	2002
Mutated gene	*Foxn1*	*Prkdc*	*Prkdc*	*Prkdc, Il-2rg*
T cell	×	×	×	×
B cell	○	×	×	×
NK cell	○	○	△	×
Engraftment of human cells				
Normal HSC	−	+	++	+++
Tumor cell	+	++	+++	++++
Success rate of PDX	Low	Low	Moderate	High

○: intact, △: deficit, ×: none. HSC: human hematopoietic cells. The Success rate of engraftment is represented by − (negative) or + (positive), with more + indicating higher possibility.

**Table 2 jpm-10-00064-t002:** Comparison of patient-derived xenograft graft survival rates based on the primary lesion site.

Tumor Type	Engraftment Rate
Melanoma	88% (n = 8)
Colorectal	85% (n = 112)
Head and neck	68% (n = 53)
Pancreatic	65% (n = 62)
Sarcoma	63% (n = 161)
Gastroesophageal	62% (n = 42)
Liver and biliary duct	54% (n = 35)
Lung	50% (n = 129)
Bladder	43% (n = 30)
Brain and neurological	40% (n = 15)
Ovarian	37% (n = 138)
Mesothelioma	36% (n = 11)
Breast	30% (n = 155)
Renal cell carcinoma	25% (n = 114)

Quoted from Ref Izumchenko et al. [57].

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
