# Peer review of "Patient-Derived Tumor Xenograft Models: Toward the Establishment of Precision Cancer Medicine"

_jpm, 2020, doi:10.3390/jpm10030064_

Round 1
Reviewer 1 Report
In this manuscript the author reviews the use of PDX. The authors described the pro and cons and concludes that these models are expected to play a major role in drug development and personalized medicine.
General comment:
The author is correct that PDX are a major step forward. However, within the PDX context, immune cell-tumor cell interaction is completely missing, whereas this has been recognized as a major component for tumor growth and response. In a review this should at least be mentioned as a possible caveat.
Considering the time required to establish a PDX and the sometimes rapid progression of patients, the time required to establish the PDX will be too long to be of use to personalize treatment. Most likely PDX-derived data combined with genetic analysis of the patients’ tumor will be more relevant. For instance: initial growth of prostate cancer PDX may take up to 9-12 month, which is clearly beyond the period that would be useful to define the best treatment modality. This issue should be highlighted in a more extensive setting. Right now the review reads as if the PDX will solve most, if not all barriers.
Specific comments:
Line 102-103: the author means to say that the human tumors will initiate a graft versus host reaction, leading to rejection in an immune competent host. This is not “heterogeneous”.
Line 107: nu/nu mice lack T cell activity, but they still are able to mount a B cell response, I.e., they are still able to mount an immune response and thus the statement is incorrect.
Moreover, strain differences have also been noted, for the establishment of PDX, this is missing in this part of the review.
Line 154 and further: patient-derived cells such as stromal cells/cancer associated fibroblast and tumor-associated macrophages. PDX models are therefore expected to be applicable to clinical and pathophysiological analyses of tumor and anti-tumor drug development because the models most closely resemble the clinical environment. PDX 157 models have already been created for many cancer types.
The text suggests that patient derived stomal cells and cancer associated fibroblasts and TAM persist after various passages. This is not true. These cells are lost and replaced by cells of mouse origin. (see also later in the manuscript, lines 296-onwards)
Line 172: NSG and NOG have the same genotype, why mention these separately?
Misspelling in table 2: Headn
Line 211: high expenses are required. The costs are high.
Line 247-248: self-replication should read self-renewal
Line 266: this sentence is confusing. The authors may mean that PDX are more informative than cell lines which have been maintained for a long periods in 2D, on a non-natural surface, which may have lead to genetic drift and selection.
Author Response
Comment 1
The author is correct that PDX are a major step forward. However, within the PDX context, immune cell-tumor cell interaction is completely missing, whereas this has been recognized as a major component for tumor growth and response. In a review this should at least be mentioned as a possible caveat.
Response: We agree with the reviewer and added the descriptions in section 3, as follows.
“In recent years, active efforts have begun to be made toward the development of new methods of cancer treatment focusing on human immunity (e.g., immune checkpoint inhibitors). The PDX model uses immunodeficient mice, i.e., mice with markedly compromised immune function. Therefore, the balance of hematopoietic and immune cells remains different from that in humans and theoretically, the immune cell-tumor cell interaction may be totally missing in this model. If these drugs are administered to immunodeficient mice, then reactions of immunocompetent cells differing from those observed in humans are anticipated. These issues require attention for the development of a patient-similar immune response PDX model.”
Comment 2
Considering the time required to establish a PDX and the sometimes rapid progression of patients, the time required to establish the PDX will be too long to be of use to personalize treatment. Most likely PDX-derived data combined with genetic analysis of the patients’ tumor will be more relevant. For instance: initial growth of prostate cancer PDX may take up to 9-12 month, which is clearly beyond the period that would be useful to define the best treatment modality. This issue should be highlighted in a more extensive setting. Right now the review reads as if the PDX will solve most, if not all barriers.
Response: We agree with the reviewer and added the descriptions in section 7, as follows.
“Meanwhile, PDX models can also take significant time to create, which may pose a challenge to patients with advanced stages of cancer. For example, growth rates of prostate cancer PDX are slow, needing many months to generate models [76]. The time to initial growth is reported to be from four up to over 12 months, and time from implantation to initial growth of secondary passage ranges from 6 to 36 weeks, partly due to the differences in androgen levels between human and mouse [26, 77]. This time-consuming process is clearly beyond the period that would be useful to define the best treatment modality.”
Comment 3
Line 102-103: the author means to say that the human tumors will initiate a graft versus host reaction, leading to rejection in an immune competent host. This is not “heterogeneous”.
Response: We revised the sentence according to the reviewer’s suggestion.
Comment 4
Line 107: nu/nu mice lack T cell activity, but they still are able to mount a B cell response, I.e., they are still able to mount an immune response and thus the statement is incorrect.
Response: We revised the sentence according to the reviewer’s suggestion.
“Dr. Rygaard (Denmark) reported a nude mouse lacking the thymus and T lymphocytes, and it displayed a defect in T-cell mediated immune responses and antibody formation that require helper T cells [34].”
Comment 5
Moreover, strain differences have also been noted, for the establishment of PDX, this is missing in this part of the review.
Response: We modified the Table 1 and added the descriptions in section 2.
|
Nude |
SCID |
NOD/SCID |
NOG |
Reporting year |
1966 |
1983 |
1995 |
2002 |
mutated gene |
Foxn1 |
Prkdc |
Prkdc |
Prkdc, Il-2rg |
T cell |
× |
× |
× |
× |
B cell |
○ |
× |
× |
× |
NK cell |
○ |
○ |
△ |
× |
Engraftment of human cells |
||||
normal HSC |
− |
+ |
++ |
+++ |
Tumor cell |
+ |
++ |
+++ |
++++ |
Success rate of PDX |
Low |
Low |
Moderate |
High |
“In addition, nude mice remain an important resource for PDX establishment because they have benefits including a relatively high engraftment ratios of gastrointestinal tumors, easy observation of subcutaneous tumors due to lack of hair, and relatively low price [31, 36, 37].”
“In essence, engraftment ratios are higher in more immunocompromised mice (Nude < SCID < NOD/ SCID < NOG) (Table 1 ) [36].”
Comment 6
Line 154 and further: patient-derived cells such as stromal cells/cancer associated fibroblast and tumor-associated macrophages. PDX models are therefore expected to be applicable to clinical and pathophysiological analyses of tumor and anti-tumor drug development because the models most closely resemble the clinical environment. PDX 157 models have already been created for many cancer types.
The text suggests that patient derived stomal cells and cancer associated fibroblasts and TAM persist after various passages. This is not true. These cells are lost and replaced by cells of mouse origin. (see also later in the manuscript, lines 296-onwards)
Response: We agree with the reviewer and added the descriptions in section 3, as follows.
“These tumors contain patient-derived cells such as stromal cells/cancer associated fibroblast and tumor-associated macrophages before they are gradually replaced by mouse cells as the passage increases. PDX models, especially in the early passages, are therefore expected to be applicable to clinical and pathophysiological analyses of tumor and anti-tumor drug development because the models most closely resemble the clinical environment.”
Comment 7
Line 172: NSG and NOG have the same genotype, why mention these separately?
Response: We agree with the reviewer and rephrased it.
Comment 8
Misspelling in table 2: Headn
Response: We corrected the misspelling.
Comment 9
Line 211: high expenses are required. The costs are high.
Response: We corrected the sentence according to the reviewer’s suggestion.
Comment 10
Line 247-248: self-replication should read self-renewal
Response: We reworded it.
Comment 11
Line 266: this sentence is confusing. The authors may mean that PDX are more informative than cell lines which have been maintained for a long periods in 2D, on a non-natural surface, which may have lead to genetic drift and selection.
Response: We agree with the reviewer and added the descriptions in section 5, as follows.
“To reveal that specific cells have the nature of stem cells in a study of cancer stem cells, it is necessary to demonstrate in vivo that a tumor composed of heterogeneous cell groups can be formed. When such an analysis is conducted, PDX are more informative than cell lines which have been maintained for a long periods in 2 dimensions, on a non-natural surface, which may have led to genetic drift and selection. [31].”
Thank you very much for your thoughtful comments.

Reviewer 2 Report
The review by Dr. Goto is a very thorough and very well-written review, which provides a new perspective to the long-going thrive for establishment of a proper model for cancer research, especially when the particular patient is taken into account. I have a couple of minor comments:
1) many times in the manuscript adjectives appear in a form without a hyphen, e.g. nonclinical, nonobese; as alternative writing might be acceptable it might be disturbing for native English readers
2) A clear Figure 1 caption is missing, a whole sentence instead
Overall, this review was very informative and a pleasure to read.
Author Response
Comment 1
many times in the manuscript adjectives appear in a form without a hyphen, e.g. nonclinical, nonobese; as alternative writing might be acceptable it might be disturbing for native English readers
Response: According to the reviewer’s suggestion, we reworded them into more common ones.
Comment 2
A clear Figure 1 caption is missing, a whole sentence instead
Response: I personally think those missing parts often happens due to the computer glitch, especially to the difference of Word software version. On my computer (MAC), the figure and its legend are fine. I, as a corresponding author, will ask the editor to correct it.